# Accuracy and repeatability of the Microsoft Azure Kinect for clinical measurement of motor function

**Johannes Bertram** [1], **Theresa Krüger**[2], **Hanna Marie Röhling**[2], **Ante Jelusic**[1], **Sebastian Mansow-Model**[2], **Roman Schniepp**[1,3], **Max Wuehr**[1], **Karen Otte** [2]*

**1** German Center for Vertigo and Balance Disorders (DSGZ), Ludwig-Maximilians-University, Munich, Germany, **2** Motognosis GmbH, Berlin, Germany, **3** Department of Neurology, Ludwig-Maximilians-University, Munich, Germany

* karen.otte@charite.de

**Data Availability Statement:** The data underlying the results presented in the study are available from the Open Science Framework (https://osf.io/z5u48/).

## Abstract

Quantitative assessment of motor function is increasingly applied in fall risk stratification, diagnosis, and disease monitoring of neuro-geriatric disorders of balance and gait. Its broad application, however, demands for low-cost and easy to use solutions that facilitate high-quality assessment outside laboratory settings. In this study, we validated in 30 healthy adults (12 female, age: 32.5 [22 – 62] years) the performance and accuracy of the latest generation of the Microsoft RGB-D camera, i.e., Azure Kinect (AK), in tracking body motion and providing estimates of clinical measures that characterise static posture, postural transitions, and locomotor function. The accuracy and repeatability of AK recordings was validated with a clinical reference standard multi-camera motion capture system (Qualisys) and compared to its predecessor Kinect version 2 (K2). Motion signal quality was evaluated by Pearson's correlation and signal-to-noise ratios while the accuracy of estimated clinical parameters was described by absolute and relative agreement based on intraclass correlation coefficients. The accuracy of AK-based body motion signals was moderate to excellent (RMSE 89 to 20 mm) and depended on the dimension of motion (highest for anterior-posterior dimension), the body region (highest for wrists and elbows, lowest for ankles and feet), and the specific motor task (highest for stand up and sit down, lowest for quiet standing). Most derived clinical parameters showed good to excellent accuracy ($r$ .84 to .99) and repeatability (ICC(1,1) .55 to .94). The overall performance and limitations of body tracking by AK were comparable to its predecessor K2 in a cohort of young healthy adults. The observed accuracy and repeatability of AK-based evaluation of motor function indicate the potential for a broad application of high-quality and long-term monitoring of balance and gait in different non-specialised environments such as medical practices, nursing homes or community centres.

## Introduction

Impairment of motor function can severely affect patients' mobility and quality of life [1]. Its clinical assessment plays a major role in the diagnosis and monitoring of disease progression

**Funding:** The work was supported by the German Federal Ministry for Education and Science (BMBF, IFB 01EO1401), the Else Kröner-Fresenius Foundation (2018EKES09 80766157) and the Federal Joint Committee (G-BA, 01NVF19030). The Motognosis GmbH provided salaries for KO, SMM, TK and HMR. The funders had no role in study design, data collection and analysis, decision to publish, or preparation of the manuscript. The specific roles of these authors are articulated in the 'author contributions' section.

**Competing interests:** I have read the journal's policy and the authors of this manuscript have the following competing interests: SMM and KO are shareholder of Motognosis GmbH and named as inventors on patent applications describing perceptive visual computing for tracking of motor dysfunction. HMR and TK are employed at Motognosis GmbH. All others report no conflict of interest. This does not alter our adherence to PLOS ONE policies on sharing data and materials.

or intervention effects in neuro-geriatric disorders of balance and gait [2, 3]. The ability to accurately and objectively assess motor function in clinical settings is essential to detect impairments at early stages of disease and to monitor a patient's condition in the long term. Currently available clinical scoring systems for motor function (e.g., UPDRS for Parkinson's disease or EDSS for Multiple Sclerosis) are limited by various factors such as bias, and a low sensitivity. This can result in low intra- and inter-rater reliability and compromise the ability to detect subtle motor impairment in particular at prodromal stages of neurodegenerative diseases [4–7].

Hence, instrument-based tools that allow to objectively assess motor impairments at a high resolution are gaining increasing importance in clinical settings [3]. Marker-based 3D motion capture systems that are based on an optical tracking of body-fixed reflective markers are currently considered the clinical reference standard in human movement analysis. However, their high cost in terms of equipment as well as required time and personnel impede their applicability in clinical care [3, 8]. Progress in computer vision has provided the means to optically detect human poses and track human motion at an increasing accuracy without the need of additional reflective markers. Such marker-free motion capture systems are more cost-effective, less time-consuming and do not require trained examiners, which makes them particularly suitable for a broad application in various in- and outpatient clinical settings [3, 9].

The Microsoft Kinect for Xbox One® (Kinect Version 2) is a marker-free motion capture system that in the past has been extensively validated and applied in (clinical) human motion analysis [10–13]. In 2019, Microsoft introduced the Microsoft Azure Kinect DK® (Kinect Version 4) as the successor to the discontinued Kinect 2. Both sensors use time of flight technology and machine learning models for spatiotemporal detection of defined anatomic landmarks. In comparison to the Kinect 2 (K2), the SDK of the Azure Kinect (AK) employs a different pose estimation algorithm for body tracking and has increased the number of provided body landmarks from previously 25 to 32. Furthermore, AK features a higher resolution of the RGB camera as well as adjustable narrow and wide field of view modes of the depth camera. The AK and its new body tracking SDK have been recently validated against marker-based motion capture systems (Vicon, Qualisys) while overground and treadmill walking in healthy adults [14–16]. According to these reports, AK yields a higher accuracy in tracking lower limb joint angles during walking and in identifying spatial step patterns (e.g., step length and width) compared to the K2. Both sensors performed equally well in temporal gait identification (e.g., step and stride time and stride time). In contrast, the spatiotemporal tracking of mid and upper body regions from AK was found to be less accurate compared to its precursor version [14].

Here, we extended these previous reports and determined the technical validity of the new AK sensor and its body tracking SDK in capturing a broad range of motion sequences that reflect motor tasks typically examined during clinical assessment (i.e., standing, sit-to-stance transitions, overground walking). For this purpose, we determined in a cohort of young healthy adults the signal accuracy as well as the accuracy and repeatability of computed clinical motion parameters for the AK in comparison to its precursor K2 and the clinical reference standard of a marker-based motion capture system (Qualisys).

## Methods

### Participants

Thirty healthy adult volunteers (12 females, age: 32.5 [22 – 62] years, height: 176.5 [156 – 190] cm, weight: 72.5 [50 – 101] kg, BMI 23.1 [15.6 – 31.0] kg/m$^2$) participated in this study, which were recruited from employees of University Hospital of Munich and their relatives. Exclusion criteria were any neurological, motor or cognitive impairment. All measurements were

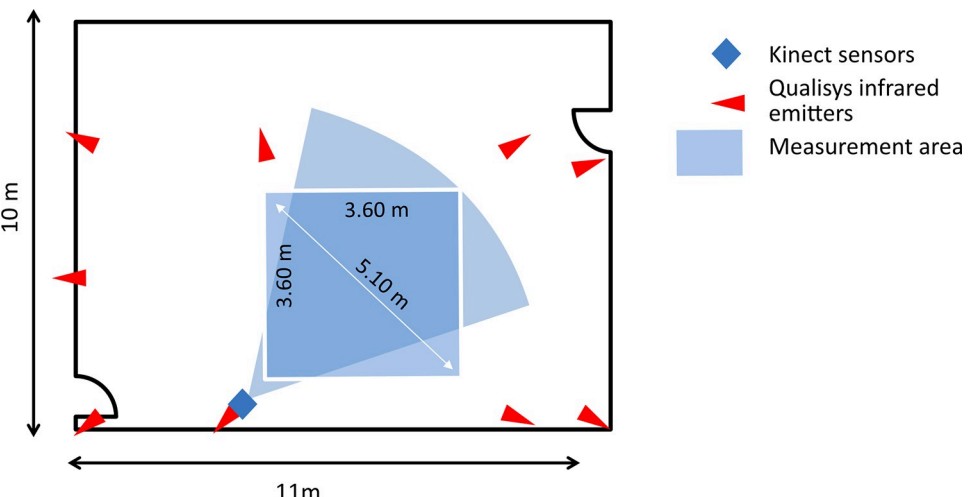

**Fig 1. Measurement setup incl. positions of Qualisys Oqus cameras and Kinect sensors.**

performed in a single recording session per individual. Each participant gave written informed consent prior to study inclusion. The ethics committee of the medical faculty of the University of Munich approved the study protocol (034-16) which was conducted in conformance with the Declaration of Helsinki.

## Experimental setup and data acquisition

All measurements took place indoors in the research facility of the department of neurology of the University Hospital of Munich. The room was dimly lit to avoid interference with natural and artificial light. A marker-based motion capture system (Qualisys AB, Sweden) consisting of nine wall-mounted Oqus cameras that covered full-body motion within an area of approximately 6x5 m was used to provide a clinical reference standard for the validation or the two Kinect sensors (see Fig 1). The recording of the Qualisys system was set to 178 Hz, with an exposure time of 75 μs and a marker threshold of 17% to allow for a compromise between the largest possible field of view and the highest framerate. Extensive spatial calibration of the system was performed each day prior to measurements covering an area of roughly 6x5 m.

   Both Kinect sensors were placed at 1.4 m height onto a tripod, mounted on top of each other. The K2 was inclined by approximately -9˚ in pitch plane to centre the recorded person. Raw data of both sensors were captured at a sampling rate of approximately 30 Hz using a customised software (Motognosis Labs v2.1.5 by Motognosis GmbH, Berlin, Germany) which utilised the Software Development SDK Version 1409 for the K2 and Version 1.4.0 with body tracking SDK 1.0.0 for the AK. Technical specifications of both Kinect sensors are given in Table 1. Data for both Kinect sensors were recorded in their respective raw formats, including infrared and depth streams (see S1 Fig).

**Table 1. Comparison technical specifications for Kinect 2 and Azure Kinect in near field of view mode (NFOV).**

|  | Kinect 2 | Azure Kinect (NFOV mode) |
| --- | --- | --- |
| RGB resolution (max) | 1920 × 1080 | 3840 × 2160 |
| Depth / IR resolution (max) | 512 × 424 | 640 × 576 |
| Depth FOV | 70˚ x 60˚ | 75˚ x 65˚ |
| Measurement Distance | 0.5–4.5 m | 0.5–5.46 m |
| Temporal resolution | 30Hz | 30Hz |
| Number of anatomical landmarks | 25 | 32 |

**Table 2. Overview of performed motor tasks.**

| Motor Task | Instruction | Movement Signals | Derived Parameters | Number of Recordings |
|---|---|---|---|---|
| Stance with closed feet and open vs. closed eyes (POCO) | Stand with closed feet and open eyes. After 20 s, an audio signal is given. Close eyes then for another 20 s. | Body sway (movement of spine base relative to closed feet position) | Pitch, roll, and 3D mean absolute angular sway speed [°/s] | 3 times |
| | | | Pitch, roll, and 3D sway deflection range [°] | |
| Stand up and Sit down (SAS) | After audio signal, stand up and wait for second audio signal to sit down. | Upper body deviation (movement of shoulder centre relative to spine base), hand range of motion in AP direction | Time needed for stand up and sit down [s] | 5 times |
| | | | Deflection range in AP direction of upper body during transitions [m] | |
| | | | Range of motion of each hand in AP direction [m] | |
| Stepping in Place (SIP) | Walk on the spot at comfortable pace for 40 s. | AP-V displacement of the knees | Step frequency (cadence) [steps/min] and step count | 3 times |
| | | | Knee range of motion in AP-V plane [cm] | |
| | | | Average step and stance time [s] | |
| Short Comfortable Speed Walk (SCSW) | After audio signal, walk directly towards the sensor at comfortable speed. | Spatial body movement (spine base movement), upper body deviation (movement of shoulder centre relative to spine base), deviation of spine base | Gait speed [m/s] and cadence [steps/min] | 5 times |
| | | | Step length [cm] and step duration [s] | |
| | | | Arm swing amplitude [°] and symmetry angle [unitless] | |

AP – Anterio-posterior, V – vertical

## Tasks/Assessments

In accordance with previous studies [10, 17, 18], each participant performed a set of motor tasks with three to five immediate, successive executions each. These tasks reflect movement sequences commonly examined during a neurological assessment of balance and gait, including a quiet stance test to measure static postural control (POCO), a stand-up and sit-down test (SAS) to measure postural transitions, as well as a stepping-in-place test (SIP) and a walking test with comfortable speed (SCSW) to measure locomotor function (see Table 2). Standardised instructions were provided prior to each task. When deviations in the performance occurred, the measurement was interrupted, discarded and the instructions were given again. Stationary tasks (POCO, SAS and SIP) were conducted at 2.5 m distance to the Kinect sensors to ensure an optimal resolution of the depth data. During the SCSW, participants were asked to start walking towards the Kinects from a position just outside the sensor range at 5.5 m distance. The SCSW task terminated automatically when participants reached 1.5 m distance to the Kinect sensors. The POCO and SIP tasks were terminated after a fixed duration of 40 s, whereas the SAS task was terminated manually. For each task, automated audio signals indicated the beginning and end of each recording.

## Movement analyses

For the Qualisys recordings, light-weight passive IR reflective markers with 19 mm diameter were placed on 36 defined anatomical landmarks [10] to capture motion of all major body joints as well as the head and the trunk. The markers were fixated on the subject's tight-fitting clothing or bare

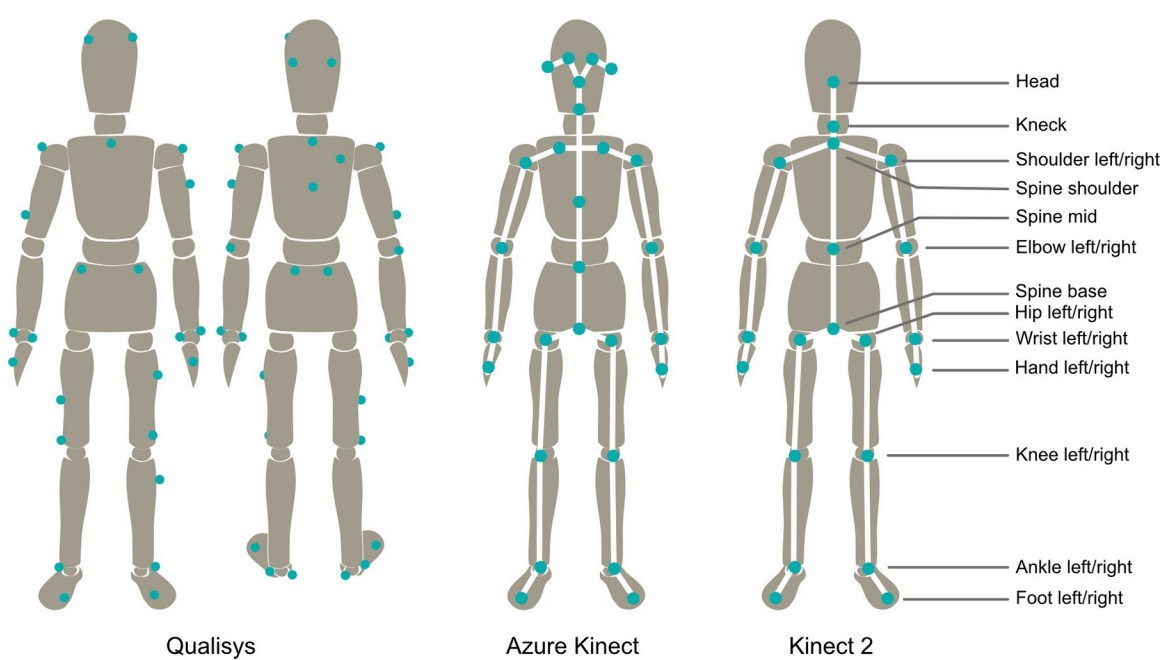

**Fig 2. Locations of Qualisys markers and artificial anatomical landmarks from the Kinect 2 and Azure Kinect.**

skin. For the Kinect sensors, body tracking was achieved by using the anatomical landmarks provided by the respective Microsoft Body Tracking SDK. Respective locations of the Qualisys markers and the artificial anatomical landmarks for both Kinect systems are shown in Fig 2.

**Spatial and temporal alignment.** Data processing was performed in MATLAB (v 2018b, The MathWorks Inc., Natick, Massachusetts, United States). To compensate for temporary loss of marker locations (flickering) in Qualisys recordings, gaps in the raw marker traces shorter than 56 ms were interpolated using polynomial spline interpolation. The anatomical landmarks of the AK and the markers from Qualisys were roughly mapped to represent landmark locations similar to K2 (as shown in the S2 Fig). This way, our results will be directly comparable to previous publications on K2 and AK accuracy.

All systems were first spatially aligned by normalizing the orientation of the coordination system (x, y and z) to describe the same dimensions. Subsequently, the tilt of the K2 sensor was compensated based on the floor normal vector provided by Kinect 2 SDK. Finally, the K2 and AK coordinate systems were rotated and translated to match the spatial orientation of Qualisys (see [10]). Next, all landmark trajectories obtained from the three systems were filtered by a 5 Hz first-order Butterworth low-pass filter. The temporal alignment between systems was achieved by down-sampling the Qualisys-derived time series to 30 Hz and then matching records by application of a temporal offset. Since some tasks like POCO yield small ranges of motion and gait tasks were non-stationary, cross-correlation alignment provided insufficient results. Therefore, offsets between systems were calculated based on system-specific time-stamps and manually adjusted for each recording. After this temporal alignment, Qualisys recordings were cut at the beginning and end to match the Kinect sensor recordings.

**Extraction of spatiotemporal parameters.** To characterise motor performance during each task, a specific set of spatiotemporal clinical parameters was calculated for each task based on the mapped and aligned motion trajectories of the anatomical landmarks. In total, 23 different parameters were computed. Postural sway during the POCO task was analysed as the maximum range and average velocity of angular sway in anterior-posterior (AP), medio-lateral

(ML) and 3D direction (the sway vector was defined as the extension of the spine base landmark relative to the mean position of both ankle landmarks) [17]. Sway parameters were calculated for the entire task duration including standing with eyes open and closed. Performance during the SAS task was quantified by the duration required to complete each transition (i.e., standing up and sitting down). Additionally, the range of movement of the shoulder spine landmark and the hands in AP direction was calculated to assess the amount of forward bending as well as potential compensatory movement strategies during both types of transitions [19]. Performance during the SIP task was characterised by a set of stepping features (i.e., cadence as the total number of steps per minute, average step and stance time) that are commonly used as surrogate markers for muscular weakness, hypokinesia or muscle fatigue [18]. In addition, the movement range was calculated as the mean amplitude of the knee landmark displacement in AP direction. Locomotor performance during the SCSW was characterised by calculating gait speed, the number of steps, the average step length, and cadence [19]. The amplitude and side asymmetry [20, 21] of arm swing during walking was calculated based on the flexion extension angle between the shoulder spine landmark and wrist landmarks.

**Statistical analysis of technical validity.** In this work, the technical validity of the body tracking capabilities of the AK were analysed as (1) the accuracy of the movement signals, (2) the accuracy of derived spatiotemporal parameters, and (3) the repeatability of spatiotemporal parameters. All statistical procedures were performed in Python 3.8 using the packages 'pandas',' scipy', 'statsmodels', 'seaborn' and 'matplotlib'.

**Accuracy of movement signals.** As described previously [10] the accuracy of movement signals was quantified by the average root-mean-square error (RMSE), the Pearson's correlation coefficients I as well as the signal-to-noise ratios (SNR) of the zero-mean shifted movement signals. Signal accuracy based on Pearson's correlation is differentiated in poor ($r < .4$), moderate ($r = .4 - .7$), good ($r = -7–0.9$) and excellent ($r > .9$) as described by Portney and Watkins [22]. SNR levels are interpreted as good (SNR > 10 dB), unclear (SNR = -10 dB – 10 dB) and bad (SNR < -10 dB) [10].

**Accuracy and repeatability of movement patterns.** The accuracy of derived spatiotemporal movement parameters was first described by providing descriptive statistics for each parameter and system as well as the absolute difference between parameters derived from each Kinect system against the Qualisys system. Absolute agreement between Qualisys against the two Kinect sensors was analysed using the intraclass correlation coefficients (ICC(A,1); two-way mixed model), whereas analysis of relative agreement that ignores potential systematic differences between systems was quantified by the Pearson's correlation coefficient. The repeatability of measures was determined by the ICC(1,1) (one-way random model) and the absolute and relative standard error of measurement (SEM).

# Results

## Interferences between Qualisys and Azure Kinect

During the system setup, we noticed a varying degree of interferences between the infrared emitters of Qualisys and the AK but not with the K2. These interferences became mainly apparent in stronger noise in the depth stream, in particular for distances above 2.5 m from the sensor as well as depth frames where large areas of depth information could not be retrieved (see black areas in S3 Fig).

The amount of impaired depth frames was inconsistent and independent from measurement setup as well as framerate of Qualisys (see S1 File). Since all stationary tasks were conducted at a distance of 2.5 m, the resulting anatomical landmark models from the AK were found to be not affected by the interference from Qualisys. Unfortunately, the stable body

tracking during the walking task was limited to distances below 3.5 m resulting in overall short gait paths (see S2 File). In comparison, the length of the gait path increased to 5.2 m after shutting down Qualisys.

From the total number of 480 performed recordings, 38 were excluded due to loss or covering of Qualisys markers (n = 19), failure of the AK SDK to extract anatomical landmarks (n = 9) and other technical errors (n = 10).

### Accuracy of movement signals

The spatial accuracy of the anatomical landmarks from K2 and AK are reported for selected landmarks as average estimates across all motor task and task repetitions in the AP (z axis), ML (x axis) and vertical dimension (y axis) (see Table 3). Task-specific results for all anatomical landmarks are given in the S1 and S2 Tables.

The overall spatial deviation of anatomical landmark position derived from the Kinect sensors to the clinical reference standard marker locations was higher for AK (RMSE > 23 mm) compared to K2 (RMSE > 20 mm), and particularly high for both sensors' estimates of appendicular landmarks, i.e., the position of the feet, ankles, and hands. Correlation results were moderate to excellent for both AK and K2 with highest agreement to the clinical reference standard in AP direction and lowest accuracy in V direction.

Task-specific SNRs (see Table 4) revealed that movement signals derived from the AK showed similar or slightly lower accuracy with respect to all landmarks compared to K2. Lowest SNR values were observed for ankle, foot, and knee landmarks in particular during stationary tasks such as POCO and SAS. Higher levels of noise in AK landmarks were additionally confirmed by visual inspection. For larger movements, however, SNR of motion signals derived from AK was in general good.

### Accuracy of spatiotemporal parameters

Based on the first measurement of each participant, descriptive statistics as well as relative and absolute agreement between K2 and AK against Qualisys are given in Table 5. Due to the very small SCSW recording area from AK (recognised walk length below 1.5 m), the recordings did not cover enough gait cycles to derive more sophisticated gait parameters. Only gait speed is reported here which is based on spine base landmarks and did not require gait cycle detection.

High to excellent relative and absolute agreement was found for spatial and temporal parameters from K2 and AK. The measurements of knee movement amplitudes during stepping in place showed only moderate absolute agreement for both sensors. Absolute agreement was overall slightly lower in AK then in K2, possibly due to small systematic differences. Blant-Alman plots for all parameters are provided in the S4 to S7 Figs.

### Repeatability of spatiotemporal parameters

Repeatability of parameters were explored based on immediate repetitions of motor tasks performances (see Table 6). We found good to excellent repeatability for all three systems for most parameters except for measures of sway range in pitch and 3D during stance and arm swing asymmetry during gait. These measures showed also large SEM above 20%. All three systems exhibited a comparable repeatability with slightly lower outcomes for the AK.

## Discussion

In this study, we evaluated the concurrent validity of the new Microsoft Azure Kinect® sensor (i.e., AK) for its application in clinical assessment of motor function. A cohort of young

Table 3. Signal accuracies for all anatomical landmarks as mean (standard deviation) of all motor tasks, participants, and repetitions.

| | RMSE [mm] | | Pearson's r in ML | | Pearson's r in V | | Pearson's r in AP | | SNR in ML | | SNR in V | | SNR AP | |
|---|---|---|---|---|---|---|---|---|---|---|---|---|---|---|
| | K2 | AK | K2 | AK | K2 | AK | K2 | AK | K2 | AK | K2 | AK | K2 | AK |
| Head | 21.15 (16.99) | 27.48 (18.06) | .69 (.45) | .52 (.54) | .86 (.17) | .56 (.44) | .99 (.05) | .95 (.11) | 4.20 (8.01) | -1.06 (7.70) | 9.46 (8.32) | 3.72 (11.94) | 24.59 (10.13) | 17.35 (8.67) |
| Neck | 33.49 (54.78) | 30.82 (32.50) | .62 (.47) | .53 (.56) | .56 (.49) | .55 (.44) | .98 (.08) | .96 (.09) | 2.31 (8.14) | -0.48 (8.22) | 3.97 (8.86) | 2.77 (12.66) | 22.72 (11.44) | 17.03 (8.77) |
| Spine shoulder | 20.30 (16.50) | 24.18 (15.76) | .66 (.48) | .58 (.53) | .80 (.21) | .59 (.41) | .99 (.05) | .98 (.05) | 2.86 (8.11) | -1.20 (8.42) | 8.62 (8.63) | 3.24 (10.47) | 24.76 (9.88) | 19.51 (7.94) |
| Shoulder L | 22.16 (16.92) | 25.07 (15.82) | .66 (.48) | .53 (.55) | .83 (.19) | .61 (.40) | .98 (.05) | .97 (.06) | 3.41 (8.49) | -0.83 (8.46) | 10.19 (9.60) | 4.42 (11.32) | 22.00 (10.15) | 17.58 (7.54) |
| Shoulder R | 22.00 (16.00) | 24.81 (16.43) | .68 (.45) | .61 (.50) | .75 (.29) | .63 (.39) | .98 (.04) | .97 (.06) | 3.33 (7.69) | -0.14 (8.20) | 9.04 (10.54) | 4.52 (11.87) | 21.43 (10.14) | 17.35 (7.44) |
| Elbow L | 29.89 (25.66) | 33.37 (30.87) | .72 (.39) | .60 (.46) | .47 (.49) | .61 (.41) | .98 (.04) | .96 (.09) | 4.97 (6.34) | 1.69 (6.20) | 4.09 (11.18) | 4.42 (10.02) | 21.43 (9.29) | 16.09 (8.43) |
| Elbow R | 30.45 (31.51) | 34.70 (34.35) | .69 (.38) | .51 (.47) | .49 (.49) | .63 (.39) | .98 (.09) | .95 (.12) | 4.24 (5.83) | 0.57 (5.73) | 4.43 (11.05) | 4.12 (10.04) | 21.76 (9.84) | 16.12 (8.51) |
| Wrist L | 30.67 (35.17) | 46.76 (53.81) | .71 (.43) | .49 (.47) | .78 (.35) | .63 (.49) | .98 (.06) | .94 (.13) | 4.84 (7.32) | -0.76 (6.33) | 8.31 (9.81) | 5.61 (10.07) | 20.00 (8.44) | 12.55 (7.60) |
| Wrist R | 28.74 (31.83) | 44.37 (48.64) | .72 (.42) | .44 (.54) | .76 (.35) | .61 (.48) | .98 (.05) | .93 (.15) | 5.04 (7.47) | -0.83 (6.73) | 7.97 (10.31) | 4.15 (10.90) | 20.82 (9.18) | 12.47 (8.38) |
| Hand L | 32.05 (38.33) | 53.31 (64.55) | .73 (.42) | .47 (.47) | .77 (.38) | .61 (.51) | .98 (.05) | .91 (.18) | 5.53 (6.99) | -1.54 (6.15) | 8.77 (9.76) | 5.16 (10.70) | 20.25 (8.25) | 11.63 (9.24) |
| Hand R | 30.43 (34.80) | 49.90 (58.33) | .72 (.42) | .50 (.48) | .78 (.34) | .58 (.50) | .98 (.06) | .90 (.23) | 5.62 (7.45) | -0.98 (6.88) | 8.44 (9.43) | 3.62 (11.00) | 20.83 (8.77) | 11.78 (9.02) |
| Spine mid | 20.21 (16.49) | 23.90 (16.51) | .65 (.49) | .59 (.53) | .77 (.24) | .62 (.39) | .99 (.05) | .98 (.03) | 2.50 (8.32) | -0.72 (8.57) | 7.60 (9.81) | 3.41 (11.77) | 23.68 (9.76) | 18.17 (7.47) |
| Spine base | 25.70 (19.09) | 27.14 (19.51) | .60 (.49) | .57 (.53) | .64 (.36) | .58 (.42) | .98 (.04) | .98 (.03) | 1.12 (7.73) | -0.60 (7.57) | 3.12 (10.51) | 0.81 (12.93) | 21.09 (9.93) | 17.51 (7.69) |
| Hip L | 30.15 (30.15) | 31.16 (30.01) | .57 (.49) | .52 (.55) | .63 (.36) | .52 (.45) | .98 (.04) | .97 (.05) | 1.23 (7.32) | -0.80 (7.34) | 3.64 (9.56) | 1.96 (11.45) | 20.79 (9.77) | 16.51 (7.78) |
| Hip R | 33.22 (36.56) | 34.10 (35.65) | .55 (.50) | .51 (.53) | .67 (.31) | .51 (.44) | .98 (.04) | .98 (.04) | 0.18 (8.32) | -1.20 (7.45) | 3.22 (9.91) | 1.21 (11.41) | 20.66 (10.02) | 16.19 (7.29) |
| Knee L | 36.91 (36.53) | 54.93 (54.68) | .66 (.49) | .38 (.50) | .30 (.49) | .43 (.48) | .90 (.17) | .74 (.33) | 2.78 (8.00) | -3.81 (6.64) | -4.93 (8.56) | -7.97 (10.38) | 12.00 (9.66) | 5.56 (10.30) |
| Knee R | 36.41 (37.20) | 57.46 (54.54) | .67 (.50) | .28 (.54) | .35 (.45) | .43 (.50) | .90 (.16) | .74 (.32) | 2.80 (8.17) | -4.46 (7.07) | -4.70 (8.02) | -8.41 (10.61) | 12.36 (9.65) | 5.45 (10.33) |
| Ankle L | 51.92 (71.21) | 77.35 (92.27) | .53 (.50) | .20 (.53) | .24 (.49) | .23 (.57) | .74 (.35) | .52 (.46) | -2.74 (9.95) | -11.31 (11.27) | -8.43 (11.54) | -12.22 (16.37) | 4.28 (14.68) | -7.37 (17.49) |
| Ankle R | 52.59 (72.65) | 83.74 (90.41) | .49 (.51) | .17 (.53) | .22 (.49) | .20 (.56) | .76 (.33) | .50 (.47) | -3.22 (9.75) | -12.86 (11.51) | -8.53 (11.37) | -13.09 (16.58) | 3.95 (15.01) | -8.55 (18.52) |
| Foot L | 64.47 (75.09) | 82.42 (100.56) | .29 (.51) | .18 (.49) | -.10 (.41) | .19 (.53) | .59 (.44) | .44 (.49) | -8.63 (11.63) | -14.50 (12.91) | -14.33 (12.83) | -15.48 (15.99) | -4.54 (19.74) | -11.26 (20.29) |
| Foot R | 64.76 (75.31) | 89.49 (97.92) | .31 (.48) | .12 (.52) | -.10 (.41) | .17 (.52) | .58 (.45) | .46 (.47) | -9.29 (12.03) | -16.54 (14.40) | -14.36 (14.00) | -16.13 (17.05) | -5.48 (21.37) | -12.05 (20.77) |

Abbr: RMSE – root mean square error, SNR – signal-to-noise ratio, ML – medio-lateral, V – vertical, AP – anterior-posterior, R –right, L – left.

healthy individuals performed different clinical tasks assessing static posture, postural transitions as well as locomotor function. The accuracy and reliability of AK for marker-less tracking of body movements during these tasks was validated against a marker-based motion capture system (i.e., Qualisys) and compared to its predecessor, the Kinect 2 (i.e., K2). Overall, AK exhibited a high accuracy and repeatability with respect to tracking of body landmarks and derived clinical outcome measures. Despite considerable differences in hardware and body

**Table 4. Signal-to-noise ratios (SNR) of Qualisys marker movements and anatomical landmark movements from Kinect v2 and Azure Kinect where higher SNR indicate higher agreement between systems.**

| | POCO | | | | | | SAS | | | | | | SIP | | | | | | SCSW | | | | | |
|---|---|---|---|---|---|---|---|---|---|---|---|---|---|---|---|---|---|---|---|---|---|---|---|---|
| | K2 ML | AK ML | K2 V | AK V | K2 AP | AK AP | K2 ML | AK ML | K2 V | AK V | K2 AP | AK AP | K2 ML | AK ML | K2 V | AK V | K2 AP | AK AP | K2 ML | AK ML | K2 V | AK V | K2 AP | AK AP |
| Head | 9 | 2 | 4 | -9 | 12 | 6 | 3 | -3 | 20 | 19 | 23 | 20 | 11 | 7 | 4 | -3 | 20 | 13 | -1 | -6 | 5 | 0 | 37 | 24 |
| Neck | 7 | 3 | -4 | -10 | 12 | 8 | 1 | -4 | 12 | 17 | 20 | 18 | 8 | 8 | -2 | -4 | 19 | 15 | -2 | -5 | 4 | 1 | 35 | 24 |
| Spine shoulder | 8 | 3 | 2 | -8 | 12 | 9 | 1 | -4 | 20 | 17 | 23 | 22 | 10 | 8 | 1 | -5 | 21 | 17 | -3 | -8 | 6 | 2 | 37 | 26 |
| Shoulder L | 9 | 3 | 3 | -7 | 11 | 8 | 1 | -5 | 23 | 20 | 19 | 19 | 11 | 10 | 4 | -2 | 16 | 15 | -3 | -6 | 5 | 1 | 35 | 24 |
| Shoulder R | 8 | 3 | 0 | -9 | 10 | 7 | 2 | -2 | 23 | 20 | 19 | 20 | 10 | 9 | 1 | -3 | 16 | 16 | -2 | -6 | 5 | 1 | 35 | 23 |
| Elbow L | 8 | 2 | -1 | -6 | 13 | 7 | 6 | 1 | 20 | 18 | 21 | 20 | 10 | 7 | -2 | -1 | 16 | 16 | -2 | -2 | -4 | 1 | 31 | 19 |
| Elbow R | 7 | 1 | -4 | -7 | 12 | 6 | 5 | 0 | 20 | 17 | 21 | 19 | 9 | 6 | -2 | 0 | 16 | 16 | -1 | -3 | -2 | 1 | 32 | 19 |
| Wrist L | 6 | -2 | -3 | -6 | 11 | 6 | 7 | 0 | 20 | 18 | 21 | 16 | 10 | 5 | 6 | 2 | 15 | 11 | -2 | -5 | 5 | 3 | 28 | 14 |
| Wrist R | 6 | -4 | -4 | -8 | 10 | 3 | 7 | 3 | 20 | 17 | 22 | 16 | 11 | 4 | 5 | 1 | 17 | 13 | -1 | -6 | 5 | 0 | 29 | 14 |
| Hand L | 6 | -4 | -5 | -9 | 10 | 0 | 9 | -1 | 19 | 18 | 22 | 18 | 11 | 4 | 7 | 3 | 16 | 13 | -1 | -5 | 8 | 2 | 28 | 12 |
| Hand R | 6 | -6 | -4 | -9 | 10 | 0 | 7 | 2 | 18 | 17 | 22 | 17 | 12 | 4 | 7 | 1 | 18 | 12 | 0 | -4 | 7 | -1 | 29 | 13 |
| Spine mid | 8 | 4 | -1 | -11 | 12 | 10 | 0 | -4 | 21 | 18 | 21 | 16 | 11 | 10 | 0 | -5 | 19 | 19 | -4 | -7 | 4 | 2 | 36 | 26 |
| Spine base | 6 | 2 | -10 | -18 | 10 | 10 | -2 | -3 | 16 | 16 | 17 | 14 | 9 | 9 | -2 | -6 | 17 | 19 | -4 | -6 | 1 | 1 | 34 | 26 |
| Hip L | 5 | 0 | -2 | -9 | 10 | 8 | -2 | -5 | 11 | 12 | 18 | 14 | 9 | 8 | 1 | -3 | 16 | 17 | -3 | -3 | 1 | 2 | 34 | 25 |
| Hip R | 5 | 0 | -1 | -9 | 10 | 8 | -6 | -5 | 9 | 10 | 17 | 14 | 9 | 7 | 1 | -3 | 15 | 16 | -3 | -3 | 1 | 1 | 34 | 24 |
| Knee L | 3 | -6 | -14 | -19 | 7 | -1 | 5 | -5 | -6 | -12 | 3 | -3 | 11 | 5 | 8 | 5 | 13 | 14 | -4 | -7 | -6 | -5 | 24 | 13 |
| Knee R | 4 | -6 | -13 | -20 | 6 | -3 | 6 | -5 | -7 | -14 | 4 | -2 | 10 | 4 | 8 | 5 | 13 | 14 | -5 | -8 | -5 | -4 | 24 | 14 |
| Ankle L | -12 | -25 | -20 | -28 | -13 | -30 | -3 | -15 | -17 | -25 | -5 | -15 | 10 | 3 | 3 | 6 | 15 | 6 | -5 | -9 | 1 | 0 | 18 | 7 |
| Ankle R | -14 | -26 | -20 | -28 | -15 | -32 | -4 | -17 | -18 | -28 | -5 | -18 | 10 | 2 | 3 | 6 | 14 | 7 | -5 | -9 | 1 | 0 | 18 | 7 |
| Foot L | -23 | -31 | -30 | -36 | -27 | -37 | -12 | -19 | -21 | -24 | -18 | -21 | 5 | 0 | 0 | 3 | 7 | 5 | -4 | -8 | -6 | -5 | 15 | 6 |
| Foot R | -26 | -36 | -34 | -38 | -32 | -38 | -12 | -22 | -20 | -25 | -18 | -23 | 4 | -2 | 0 | 2 | 7 | 5 | -4 | -8 | -6 | -4 | 16 | 5 |
| Total | 2 | -6 | -8 | -15 | 4 | -2 | 1 | -5 | 9 | 6 | 13 | 9 | 9 | 6 | 3 | 0 | 16 | 13 | -3 | -6 | 1 | 0 | 29 | 17 |

Abbr: ML – medio-lateral, V – vertical, AP – anterio-posterior, R – right, L – left.

tracking algorithms between AK and K2, both sensors yielded similar performance and limitations in body tracking capability. In the following, we will discuss these findings with respect to the accuracy and limitations of AK for clinical motion tracking and potential fields of clinical applications.

When compared to the Qualisys clinical reference system, AK-derived movement signals yielded a moderate to high accuracy of body motion signals that depended on (1) the dimension of movement, (2) the body landmark location, and (3) the clinical motor task. Accordingly, motion signal accuracies were highest in the AP dimension and lowest for movements along the vertical dimension. Tracking of body landmarks was most precise for the head, trunk, and upper extremities but more or less compromised with respect to the lower extremities, in particular with respect to movements of the ankles and feet. Moreover, tracking accuracy depended on the specific motor task and exhibited limited precision with a high signal-to-noise ratio (SNR) in the case of stationary landmarks, e.g., tracking of lower extremities during quiet stance. This limitation has been previously noted and interpreted as a specific difficulty of marker-less tracking approaches to differentiate the feet from floor in the case of close proximity [10, 22]. Also, AK seems to show slightly higher levels of noise in vertical direction, especially during quiet stance. However, irrespective of body region and movement

**Table 5. Accuracy of parameters using only the first available measurement per person.**

| | Q mean (SD) | K2 mean (SD) | AK mean (SD) | K2 vs Q Diff | AK vs Q Diff | K2 vs Q Pearson's r | AK vs Q Pearson's r | K2 vs Q ICC (A,1) | AK vs Q ICC (A,1) |
|---|---|---|---|---|---|---|---|---|---|
| **Balance (POCO)** | | | | | | | | | |
| **3D sway speed [°/s]** | 0.23 (0.09) | 0.23 (0.10) | 0.29 (0.11) | 0 | 0.07 | .92 | .93 | .91 | .75 |
| **Roll sway speed [°/s]** | 0.12 (0.08) | 0.13 (0.09) | 0.17 (0.10) | 0.01 | 0.05 | .89 | .88 | .88 | .74 |
| **Pitch sway speed [°/s]** | 0.17 (0.05) | 0.16 (0.06) | 0.19 (0.06) | -0.01 | 0.03 | .94 | .91 | .91 | .76 |
| **3D sway range [°]** | 1.28 (0.43) | 1.41 (0.52) | 1.40 (0.53) | 0.13 | 0.18 | .93 | .90 | .89 | .84 |
| **Roll sway range [°]** | 0.75 (0.41) | 0.90 (0.58) | 1.08 (0.54) | 0.15 | 0.30 | .91 | .84 | .83 | .68 |
| **Pitch sway range [°]** | 1.33 (0.43) | 1.45 (0.51) | 1.43 (0.55) | 0.12 | 0.17 | .96 | .91 | .92 | .85 |
| **Stand up and sit down (SAS)** | | | | | | | | | |
| **Transition time (up) [s]** | 1.47 (0.23) | 1.44 (0.22) | 1.45 (0.20) | -0.03 | 0 | .99 | .97 | .98 | .97 |
| **AP deflection range (up) [m]** | 0.44 (0.10) | 0.43 (0.10) | 0.43 (0.09) | -0.01 | 0 | .99 | .98 | .99 | .98 |
| **Hand AP range (up) [m]** | 0.39 (0.09) | 0.39 (0.09) | 0.38 (0.09) | 0 | -0.01 | .95 | .91 | .956 | .91 |
| **Transition time (down) [s]** | 1.59 (0.22) | 1.34 (0.25) | 1.51 (0.24) | -0.25 | -0.06 | .92 | .97 | .57 | .94 |
| **AP deflection range (down) [m]** | 0.45 (0.10) | 0.38 (0.13) | 0.44 (0.10) | -0.07 | 0 | .94 | .98 | .77 | .98 |
| **Hand AP range (down) [m]** | 0.39 (0.09) | 0.30 (0.11) | 0.37 (0.08) | -0.08 | -0.02 | .84 | .95 | .60 | .91 |
| **Stepping in Place (SIP)** | | | | | | | | | |
| **Knee amplitude [cm]** | 0.24 (0.06) | 0.18 (0.05) | 0.20 (0.05) | -0.06 | -0.03 | .97 | .77 | .63 | .63 |
| **Stepping cadence [steps/min]** | 88.2 (14.5) | 88.3 (14.5) | 93.0 (15.5) | 0.12 | 2.45 | 1.0 | .87 | 1.0 | .86 |
| **Step duration (s)** | 0.86 (0.10) | 0.85 (0.10) | 0.86 (0.10) | -0.01 | 0.01 | .97 | .66 | .97 | .66 |
| **Stance duration (s)** | 0.47 (0.15) | 0.48 (0.14) | 0.41 (0.15) | 0.01 | -0.04 | .99 | .95 | .99 | .93 |
| **Gait (SCSW)** | | | | | | | | | |
| **Gait speed (m/s)** | 1.11 (0.19) | 1.10 (0.18) | 0.99 (0.17) | -0.01 | -0.18 | .99 | .85 | .99 | .53 |
| **Step length (cm)** | 64.89 (8.14) | 65.02 (7.89) | n.a. | 0.07 | n.a. | .97 | n.a. | .97 | n.a. |
| **Gait cadence (steps/min)** | 113.9 (10.5) | 117.6 (13.9) | n.a. | 3.24 | n.a. | .96 | n.a. | .89 | n.a. |
| **Step duration (s)** | 0.51 (0.04) | 0.50 (0.05) | n.a. | -0.01 | n.a. | .96 | n.a. | .95 | n.a. |
| **Arm angular amplitude (°)** | 25.25 (8.39) | 25.38 (9.67) | n.a. | 0.09 | n.a. | .99 | n.a. | .98 | n.a. |
| **Arm symmetry angle [n.u.]** | 0.20 (0.17) | 0.20 (0.17) | n.a. | 0 | n.a. | .93 | n.a. | .94 | n.a. |

n.a. – not available due to too few gait cycles.

dimension, we observed that AK yielded excellent accuracy in the case of large amplitude body motions. The observed dependency of AK tracking accuracy with respect to movement dimension and body location concurs with previous reports that focused on AK-based assessment of treadmill locomotion [14, 15].

For each clinical motor task, we calculated a set of common spatiotemporal outcome measures based on the recorded body motion patterns. Overall, the AK-derived clinical outcome measures yielded a good to excellent relative and absolute agreement with the Qualisys clinical reference standard. Lowest, but still moderate, agreement was found for measures of static postural sway and timing while stepping in place. Differences in static sway measures (in particular in ML dimension) could be caused by a systematic spatial offset of the underlying spine base landmark between AK and Qualisys. The moderate agreement with respect to step timing

**Table 6. Repeatability of spatiotemporal parameters derived from 3 to 5 repeated measurements as intra-class correlation coefficient ICC(1,1) and standard error of measurement (SEM %).**

| | Q Mean (SD) | K2 Mean (SD) | AK Mean (SD) | Q ICC (1,1) | K2 ICC (1,1) | AK ICC (1,1) | Q SEM % | K2 SEM % | AK SEM % |
|---|---|---|---|---|---|---|---|---|---|
| **Balance (POCO) (rep = 3)** | | | | | | | | | |
| **3D sway speed [°/s]** | 0.23 (0.09) | 0.23 (0.11) | 0.29 (0.11) | .90 | .93 | .89 | 11.8 | 12.7 | 12.0 |
| **Roll sway speed [°/s]** | 0.12 (0.08) | 0.13 (0.09) | 0.17 (0.10) | .95 | .96 | .87 | 14.7 | 14.7 | 19.4 |
| **Pitch sway speed [°/s]** | 0.16 (0.05) | 0.15 (0.06) | 0.19 (0.06) | .76 | .81 | .82 | 14.9 | 17.0 | 13.0 |
| **3D sway range [°]** | 1.22 (0.45) | 1.32 (0.51) | 1.40 (0.53) | .50 | .63 | .58 | 26.0 | 23.8 | 24.5 |
| **Roll sway range [°]** | 0.77 (0.43) | 0.95 (0.58) | 1.08 (0.53) | .87 | .90 | .78 | 20.1 | 19.9 | 23.5 |
| **Pitch sway range [°]** | 1.26 (0.48) | 1.35 (0.56) | 1.43 (0.55) | .55 | .60 | .55 | 25.3 | 26.5 | 25.8 |
| **Stand up and sit down (SAS) (rep = 5)** | | | | | | | | | |
| **Transition time (up) [s]** | 1.46 (0.22) | 1.43 (0.21) | 1.45 (0.20) | .80 | .80 | .75 | 6.7 | 6.5 | 6.9 |
| **AP deflection range (up) [m]** | 0.43 (0.09) | 0.42 (0.09) | 0.43 (0.09) | .89 | .89 | .88 | 7.0 | 7.4 | 7.3 |
| **Hand AP range (up) [m]** | 0.40 (0.08) | 0.39 (0.09) | 0.38 (0.09) | .63 | .69 | .76 | 12.8 | 12.2 | 11.2 |
| **Transition time (down) [s]** | 1.57 (0.24) | 1.37 (0.30) | 1.52 (0.24) | .78 | .82 | .81 | 7.1 | 9.2 | 6.9 |
| **AP deflection range (down) [m]** | 0.44 (0.09) | 0.39 (0.12) | 0.44 (0.09) | .86 | .82 | .85 | 7.9 | 12.6 | 8.3 |
| **Hand AP range (down) [m]** | 0.39 (0.08) | 0.32 (0.10) | 0.37 (0.08) | .66 | .70 | .71 | 12.5 | 16.4 | 11.7 |
| **Stepping in Place (SIP) (rep = 3)** | | | | | | | | | |
| **Knee amplitude [cm]** | 0.24 (0.06) | 0.18 (0.05) | 0.20 (0.05) | .94 | .96 | .94 | 5.9 | 5.8 | 5.8 |
| **Stepping cadence [steps/min]** | 90.7 (13.8) | 90.4 (14.2) | 93.0 (15.5) | .88 | .94 | .94 | 5.2 | 3.9 | 4.3 |
| **Step duration (s)** | 0.85 (0.09) | 0.84 (0.09) | 0.86 (0.11) | .91 | .93 | .91 | 3.2 | 3.0 | 3.6 |
| **Stance duration (s)** | 0.45 (0.14) | 0.46 (0.14) | 0.41 (0.15) | .86 | .91 | .93 | 12.0 | 8.7 | 10.1 |
| **Gait (SCSW) (rep = 5)** | | | | | | | | | |
| **Gait speed (m/s)** | 1.15 (0.21) | 1.14 (0.20) | 0.99 (0.17) | .89 | .84 | .81 | 6.1 | 7.0 | 7.4 |
| **Step length (cm)** | 66.05 (8.75) | 65.60 (8.08) | n.a. | .90 | .90 | n.a. | 4.1 | 4.0 | n.a. |
| **Gait cadence (steps/min)** | 116.2 (11.5) | 118.8 (12.7) | n.a. | .82 | .83 | n.a. | 4.2 | 4.5 | n.a. |
| **Step duration (s)** | 0.50 (0.05) | 0.50 (0.05) | n.a. | .85 | .79 | n.a. | 3.7 | 4.4 | n.a. |
| **Arm angular amplitude (°)** | 25.66 (8.9) | 26.46 (10.7) | n.a. | .92 | .95 | n.a. | 9.7 | 9.1 | n.a. |
| **Arm symmetry angle [n.u.]** | 0.17 (0.14) | 0.18 (0.15) | n.a. | .59 | .65 | n.a. | 52.7 | 49.0 | n.a. |

n.a. – not available due to too few gait cycles.

is likely due to a discrepancy in the recognised number of steps between AK and Qualisys. Furthermore, almost all AK-derived spatiotemporal outcome measures showed a good to excellent consistency between repeated assessments of the same clinical motor task. Accuracy and repeatability measures presented for the K2 concur with previous publications [9–13].

Three studies recently demonstrated a high validity of the AK sensor in estimating spatiotemporal gait measures during overground and treadmill locomotion [14–16]. Our current observations extend these previous findings and demonstrate that the AK provides valid and reliable estimates of spatiotemporal outcome measures in young healthy adults that are commonly used in the clinical evaluation motor dysfunction. Quantitative assessments of static posture, postural transition, and locomotor function have been frequently shown to entail important information for a personalised fall risk estimation in neuro-geriatric patients and to reflect disease severity and progression of various neurodegenerative disorders [8, 23–25]. Hence, analogous to its predecessor K2 [18], the comparatively low equipment and personnel costs of AK may facilitate a broad application of reliable and high-resolution balance and gait assessment in various clinical and non-clinical environments such as outpatient clinics, medical practices, nursing homes or community centres. Furthermore, the observed consistency of AK-derived measures between repeated assessments may in particular facilitate an objective

monitoring of subtle disease or age related alterations of motor function in the long term. However, since the present validation experiments solely focused on a young, healthy population, subsequent studies are required to assess the validity and reliability of AK-based assessment of motor function in more heterogeneous study populations that are expected to show more variable movement patterns (eg, elderly people or persons with motoric impairments).

Compared to its predecessor K2, which is no longer manufactured, the AK underwent major developments in particular with respect to its optical hardware and the utilised body tracking approach. While body pose tracking of the K2 is based on a random forest network trained on the sensor's depth images [26], the AK features both an improved resolution of the integrated depth camera and a refined body tracking approach that utilises convolutional neural networks based on recent developments in deep learning. These hardware and software changes could be expected to yield a more accurate body tracking performance. In line with these assumptions, AK tracking performance during treadmill locomotion was previously shown to surpass its predecessor K2, even though overall performance differences were only moderate to marginal [14, 15]. In contrast, we found an overall similar performance between AK and K2 sensors, both with respect to the accuracy of landmark motion signals and the validity of derived clinical outcome measures. Moreover, tracking performance of AK was found to be even inferior to K2 at larger distances from the sensor. The latter can be explained by the presence of interferences between the clinical reference standard Qualisys motion capture system and the Kinect sensors that both operate in the overlapping regions of the IR spectrum. These interferences have been noted previously [27]: while Qualisys recordings appear to be not affected, the depth recordings of Kinect sensors becomes noticeably distorted at larger distances from the sensor. However, except for the gait task (SCSW), all other clinical tasks were performed in close proximity to the Kinect sensors and should therefore not be affected by Qualisys-induced distortions in the depth stream.

In conclusion, the presented validation experiments in young, healthy individuals demonstrate that AK is able to accurately monitor body movements and to provide reliable estimates of gait and balance capacity during a variety of motor tasks commonly performed in clinical assessment. Hence, AK represents a valid but seemingly not superior alternative for its predecessor sensor K2, which is no longer manufactured. Further studies are required to confirm the current observations for the application of AK-based motion analysis in different clinical cohorts.

## Supporting information

**S1 File.**
(XLSX)

**S2 File.**
(XLSX)

**S1 Fig. Comparison of Kinect version 2 and Azure Kinect infrared and depth images.** The recording angle slightly differs between both sensors, resulting in more pronounced distortions of the walls in Azure Kinect images.
(TIF)

**S2 Fig. Mapping of anatomical Landmarks and their names from the Kinect Version 2.**
(TIF)

**S3 Fig. Sequence of Kinect 4 depth frames where the lower left frame shows depth interferences as larger black areas in comparison to the previous and following frames.**
(TIF)

**S4 Fig. Bland-Altman plots comparing accuracy of measures from the Balance task (POCO) for Kinect Version 2 (blue – round) and Azure Kinect (cross – orange).** Outliers above and below the Limit of Agreement (LOA) are marked red.
(TIF)

**S5 Fig. Bland-Altman plots comparing accuracy of measures from the Stand up and sit down task (SAS) for Kinect Version 2 (blue – round) and Azure Kinect (cross – orange).** Outliers above and below the Limit of Agreement (LOA) are marked red.
(TIF)

**S6 Fig. Bland-Altman plots comparing accuracy of measures from the Stepping in Place task (SIP) for Kinect Version 2 (blue – round) and Azure Kinect (cross – orange).** Outliers above and below the Limit of Agreement (LOA) are marked red.
(TIF)

**S7 Fig. Bland-Altman plots comparing accuracy of measures from the Short comfortable speed walk task (SCSW) for Kinect Version 2 (blue – round) and Azure Kinect (cross – orange).** Outliers above and below the Limit of Agreement (LOA) are marked red.
(TIF)

**S1 Video. Video of the Azure Kinect Depth data from a balance task recording showing inferences by the Qualisys infrared emitters as flickers of larger black areas for some frames.**
(MP4)

**S2 Video. Video of the Azure Kinect Depth data from a short gait task recording showing inferences by the Qualisys infrared emitters as flickers of larger black areas for some frames.**
(MP4)

**S1 Table.**
(XLSX)

**S2 Table.**
(XLSX)

## Author Contributions

**Conceptualization:** Sebastian Mansow-Model, Roman Schniepp, Max Wuehr, Karen Otte.

**Data curation:** Johannes Bertram, Theresa Krüger, Hanna Marie Röhling, Ante Jelusic, Karen Otte.

**Formal analysis:** Johannes Bertram, Theresa Krüger, Hanna Marie Röhling, Karen Otte.

**Funding acquisition:** Roman Schniepp, Max Wuehr.

**Investigation:** Johannes Bertram, Theresa Krüger, Karen Otte.

**Methodology:** Johannes Bertram, Hanna Marie Röhling, Roman Schniepp, Karen Otte.

**Project administration:** Roman Schniepp, Max Wuehr, Karen Otte.

**Resources:** Roman Schniepp, Max Wuehr, Karen Otte.

**Software:** Hanna Marie Röhling, Karen Otte.

**Supervision:** Roman Schniepp, Max Wuehr, Karen Otte.

**Validation:** Johannes Bertram, Theresa Krüger, Karen Otte.

**Visualization:** Karen Otte.

**Writing – original draft:** Johannes Bertram, Theresa Krüger, Max Wuehr, Karen Otte.

**Writing – review & editing:** Johannes Bertram, Theresa Krüger, Hanna Marie Röhling, Ante Jelusic, Sebastian Mansow-Model, Roman Schniepp, Max Wuehr, Karen Otte.

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
