## [Decision Letter · Decision Letter 0]

29 Sep 2022

PONE-D-22-18764Accuracy and repeatability of the Microsoft Azure Kinect for clinical measurement of motor functionPLOS ONE

Dear Dr. Otte,

Thank you for submitting your manuscript to PLOS ONE. After careful consideration, we feel that it has merit but does not fully meet PLOS ONE’s publication criteria as it currently stands. Therefore, we invite you to submit a revised version of the manuscript that addresses the points raised during the review process.

You will see from the Reviewer's comments, that although they were both favourable in terms of the idea, they have raised some concerns with regards to the statistical analysis used and the limitations in generalisability of the study. Therefore, I would strongly encourage you to carefully consider the comments and address them appropriately.  In particular:Consider removing the information relating to the discontinued productAdd Limits of Agreement in your analysis, as it would not only substantiate it further but also allow the reader a more informed decision on the agreement of the new deviceInclude the limitations identified as part of your discussion

We look forward to receiving your revised manuscript.

Kind regards,

Theodoros M. Bampouras

Academic Editor

PLOS ONE

Journal Requirements:

“The work was supported by the German Federal Ministry for Education and Science (BMBF, IFB 01EO1401), the Else Kröner-Fresenius Foundation (2018EKES09 80766157) and the Federal Joint Committee (G-BA, 01NVF19030). The Motognosis GmbH provided salaries for KO, SMM, TK and HMR.”

“I have read the journal's policy and the authors of this manuscript have the following competing interests: SMM and KO are are shareholder of Motognosis GmbH and named as inventors on patent applications describing perceptive visual computing for tracking of motor dysfunction. HMR and TK are employed at Motognosis GmbH. All others report no conflict of interest.”

4. We note that you have stated that you will provide repository information for your data at acceptance. Should your manuscript be accepted for publication, we will hold it until you provide the relevant accession numbers or DOIs necessary to access your data. If you wish to make changes to your Data Availability statement, please describe these changes in your cover letter and we will update your Data Availability statement to reflect the information you provide

7. We note you have included a table to which you do not refer in the text of your manuscript. Please ensure that you refer to Tables 3 and 6  in your text; if accepted, production will need this reference to link the reader to the Table.

8. Please include a copy of Table 5 which you refer to in your text on page 8.

Reviewers' comments:

Reviewer's Responses to Questions

**Comments to the Author**

1. Is the manuscript technically sound, and do the data support the conclusions?

Reviewer #1: Yes

Reviewer #2: Yes

2. Has the statistical analysis been performed appropriately and rigorously? 

Reviewer #1: Yes

Reviewer #2: Yes

3. Have the authors made all data underlying the findings in their manuscript fully available?

Reviewer #1: Yes

Reviewer #2: Yes

4. Is the manuscript presented in an intelligible fashion and written in standard English?

Reviewer #1: Yes

Reviewer #2: Yes

5. Review Comments to the Author

Reviewer #1: Dear Authors.

The work is very interesting and supports the results of others previously published with the Kinect One camera that this system is an easy solution for the evaluation of movement. The only problem is in relation to the sample size and the non-evaluation by age range.

The authors should explain the limitations of the sample and the age of the participants that prevent them from demonstrating the strength of the results.

Reviewer #2: This manuscript is well done and achieves its aim to assess the agreement between a marker system (the ‘gold’ standard Qualysis) against a newer marker-less system, and their respective reliability.

I have several observations which should be strongly considered before publication.

1. I am uncertain of the role of the K2 in this study. There is no convincing argument for including this device and comparing it to the others, especially when (as the authors state in at least 2 sentences) the K2 is no longer manufactured. If the K2 has never been compared to a 'gold standard' (which is implied on line 61,) it's a bit late!. Having these data also complicates the agreement and reliability analysis.

2. The statistical analysis is well done and comprehensive. However, when assessing the level of agreement between the systems (and hence the newer system’s (i.e. AK) validity against the ‘gold’ standard) the authors have opted for ICC (A,1). I strongly expect most readers will be also expecting to see the more familiar Bland and Altman plots to assess levels of agreement. Even though the ICC (A,1) does this, the visual effect of the Bland and Altman plots are important.

3. The second piece of advice surrounds the phrase ‘gold standard’. A more appropriate term, in this study using these systems, would be ‘clinical reference standard’. This means that it is the best available method, rather than the perfect method which the phrase ‘gold standard’ often implies.

4. Technical points. Figures 2 & 4 were unhelpful. Not sure what readers are supposed to see in this. This was even after I downloaded the tiff files.

Typographical errors

Line 57 spelling ‘these’

Line 57-62: it makes more sense to have the verbs in the past tense (i.e. extended, determined)

Line 68-69: This makes more sense to say that these were “exclusion criteria”

Line 76: “day- and artificial light“ ; delete the hypen

Line 114: insert ‘were terminated’

Line 303 “subtle disease-“ delete the hyphen

Line 307: it’s better to write more simply: “which is no longer manufactured”; I would repeat this version in the conclusion on line 328.

Line 325: The word ‘current’ here is actually a little confusing as at first glance it seems it should be ‘concurrent validation experiments’ (which I presume it is not). It would be better to write: “ In conclusion, the present validation experiments….”

6. PLOS authors have the option to publish the peer review history of their article (what does this mean?). If published, this will include your full peer review and any attached files.

Reviewer #1: No

Reviewer #2: No

---

## [Author Response · Author response to Decision Letter 0]

5 Nov 2022

We would like to thank both reviewers for their time and valuable feedback. Please find attached our detailed responses to you feedback below. 

Feedback from Reviewer #1: Dear Authors.

The work is very interesting and supports the results of others previously published with the Kinect One camera that this system is an easy solution for the evaluation of movement. The only problem is in relation to the sample size and the non-evaluation by age range.

The authors should explain the limitations of the sample and the age of the participants that prevent them from demonstrating the strength of the results.

Answer to the feedback from the authors:

Throughout the manuscript, we strengthened the fact that the cohort consists of young healthy participants and advanced the statement regarding the study limitations. (see line 298ff).

 

Feedback from Reviewer #2: This manuscript is well done and achieves its aim to assess the agreement between a marker system (the ‘gold’ standard Qualysis) against a newer marker-less system, and their respective reliability.

I have several observations which should be strongly considered before publication.

1. I am uncertain of the role of the K2 in this study. There is no convincing argument for including this device and comparing it to the others, especially when (as the authors state in at least 2 sentences) the K2 is no longer manufactured. If the K2 has never been compared to a 'gold standard' (which is implied on line 61,) it's a bit late!. Having these data also complicates the agreement and reliability analysis. 

Answer to the feedback from the authors:

Thank you for your feedback in this regard. Our wording in line 61 seems to have been unclear, so we have rewritten the sentence. The Kinect version 2 (K2) has been used for a variety of clinical trials in recent years and, despite being discontinued, is still very much in use for research purposes as recent publications show (eg, Ma et al. Kinect V2-Based Gait Analysis for Children with Cerebral Palsy: Validity and Reliability of Spatial Margin of Stability and Spatiotemporal Variables. Sensors (Basel). 2021 Mar 17;21(6):2104). The camera and its pose estimation software was extensively investigated regarding their technical and clinical feasibility previously. And in some clinical studies, the K2 was even used as the only measurement device for motor function, without any additional clinical-reference system. The new generation of sensors has just entered this testing phase regarding its usability, which is why we think it is of importance and an actual strength of our manuscript to not only provide marker-based reference measures but also results from the K2. In this way, readers and researchers can compare their existing study results based on the K2 and can estimate potential differences to the new sensor system. This is in our opinion one of the important information required for switching to the K4.

2. The statistical analysis is well done and comprehensive. However, when assessing the level of agreement between the systems (and hence the newer system’s (i.e. AK) validity against the ‘gold’ standard) the authors have opted for ICC (A,1). I strongly expect most readers will be also expecting to see the more familiar Bland and Altman plots to assess levels of agreement. Even though the ICC (A,1) does this, the visual effect of the Bland and Altman plots are important.

Answer to the feedback from the authors:

We created the Bland-Altman plots and added them as supplementary figures 4 to 7.

3. The second piece of advice surrounds the phrase ‘gold standard’. A more appropriate term, in this study using these systems, would be ‘clinical reference standard’. This means that it is the best available method, rather than the perfect method which the phrase ‘gold standard’ often implies.

Answer to the feedback from the authors:

We changed the phrasing throughout the whole manuscript. 

4. Technical points. Figures 2 & 4 were unhelpful. Not sure what readers are supposed to see in this. This was even after I downloaded the tiff files.

Answer to the feedback from the authors:

We moved these figures to the supplementary material. Our reasoning against a deletion of these figures is, that for researchers familiar with these systems the information is quite valuable. Either to compare the image resolution as well as to get an idea how these inferences would look like in the recorded depth data.

Typographical errors

Line 57 spelling ‘these’

Line 57-62: it makes more sense to have the verbs in the past tense (i.e. extended, determined)

Line 68-69: This makes more sense to say that these were “exclusion criteria”

Line 76: “day- and artificial light“ ; delete the hypen

Line 114: insert ‘were terminated’

Line 303 “subtle disease-“ delete the hyphen

Line 307: it’s better to write more simply: “which is no longer manufactured”; I would repeat this version in the conclusion on line 328.

Line 325: The word ‘current’ here is actually a little confusing as at first glance it seems it should be ‘concurrent validation experiments’ (which I presume it is not). It would be better to write: “ In conclusion, the present validation experiments….”

Answer to the feedback from the authors:

We fixed all typographical errors provided. Thank you again very much for your support and valuable feedback to our manuscript.

---

## [Decision Letter · Decision Letter 1]

13 Dec 2022

Accuracy and Repeatability of the Microsoft Azure Kinect for clinical Measurement of motor Function

PONE-D-22-18764R1

Dear Dr. Otte,

We’re pleased to inform you that your manuscript has been judged scientifically suitable for publication and will be formally accepted for publication once it meets all outstanding technical requirements.

Kind regards,

Theodoros M. Bampouras

Academic Editor

PLOS ONE

Additional Editor Comments: Please note the issues raised by Reviewer 2 re the image files and ability to play the files. Once in production, updated files may be required.

Reviewers' comments:

Reviewer's Responses to Questions

**Comments to the Author**

1. If the authors have adequately addressed your comments raised in a previous round of review and you feel that this manuscript is now acceptable for publication, you may indicate that here to bypass the “Comments to the Author” section, enter your conflict of interest statement in the “Confidential to Editor” section, and submit your "Accept" recommendation.

Reviewer #1: All comments have been addressed

Reviewer #2: (No Response)

2. Is the manuscript technically sound, and do the data support the conclusions?

Reviewer #1: Yes

Reviewer #2: Yes

3. Has the statistical analysis been performed appropriately and rigorously? 

Reviewer #1: Yes

Reviewer #2: Yes

4. Have the authors made all data underlying the findings in their manuscript fully available?

Reviewer #1: Yes

Reviewer #2: Yes

5. Is the manuscript presented in an intelligible fashion and written in standard English?

Reviewer #1: Yes

Reviewer #2: Yes

6. Review Comments to the Author

Reviewer #1: Dear Authors

Thank you for responding appropriately to the suggestions made on the manuscript. I do not require any further clarification

Reviewer #2: All comments have been addressed.

Bland and Altman plots have been added and are now viewable.

However, the quality of supplementary files S1,2,3 is still very poor. I don't know if this will improve in the finalised published form.

7. PLOS authors have the option to publish the peer review history of their article (what does this mean?). If published, this will include your full peer review and any attached files.

Reviewer #1: **Yes: **GRACIA CASTRO-LUNA

Reviewer #2: No

---

## [Editor Report · Acceptance letter]

12 Jan 2023

PONE-D-22-18764R1 

Accuracy and Repeatability of the Microsoft Azure Kinect for clinical Measurement of motor Function 

Dear Dr. Otte:

I'm pleased to inform you that your manuscript has been deemed suitable for publication in PLOS ONE. Congratulations! Your manuscript is now with our production department. 

Kind regards, 

on behalf of

Dr. Theodoros M. Bampouras 

Academic Editor

PLOS ONE